# Protocol for a scoping review exploring the use of patient-reported outcomes in adult social care

Sarah E Hughes ,[1,2,3] Olalekan Lee Aiyegbusi,[1,2,3,4,5] Daniel S Lasserson,[6]
Philip Collis,[1] Samantha Cruz Rivera ,[1,2,7] Christel McMullan ,[1,2]
Grace M Turner,[1,2] Jon Glasby,[8] Melanie Calvert [1,2,3,4,7,9]

For numbered affiliations see end of article.

**Correspondence to**
Sarah E Hughes;
s.e.hughes@bham.ac.uk

## ABSTRACT

**Introduction** Patient-reported outcomes (PROs) are measures of a person's own views of their health, functioning and quality of life. They are typically assessed using validated, self-completed questionnaires known as patient-reported outcome measures (PROMs). PROMs are used in healthcare settings to support care planning, clinical decision-making, patient–practitioner communication and quality improvement. PROMs have a potential role in the delivery of social care where people often have multiple and complex long-term health conditions. However, the use of PROMs in this context is currently unclear. The objective of this scoping review is to explore the evidence relating to the use of PROMs in adult social care.

**Methods and analyses** The electronic databases Medline (Ovid), PsychInfo (Ovid), ASSIA (ProQuest), Social Care Online (SCIE), Web of Science and EMBASE (Ovid) were searched on 29 September 2020 to identify eligible studies and other publically available documents published since 2010. A grey literature search and hand searching of citations and reference lists of the included studies will also be undertaken. No restrictions on study design or language of publication will be applied. Screening and data extraction will be completed independently by two reviewers. Quality appraisal of the included documents will use the Critical Appraisal Skills Programme and AACODS (Authority, Accuracy, Coverage, Objectivity, Date, Significance) checklists. A customised data charting table will be used for data extraction, with analysis of qualitative data using the framework method. The review findings will be presented as tables and in a narrative summary.

**Ethics and dissemination** Ethical review is not required as scoping reviews are a form of secondary data analysis that synthesise data from publically available sources. Review findings will be shared with service users and other relevant stakeholders and disseminated through a peer-reviewed publication and conference presentations. This protocol is registered on the Open Science Framework (www.osf.io).

## Strengths and limitations of this study

► This scoping review will follow the Preferred Reporting Items for Systematic Reviews and Meta-analyses (PRISMA) guidelines for the conduct of scoping reviews, ensuring transparency and rigour.

► Service user and community involvement in the design and conduct of the review will ensure that the findings are supported by stakeholder experiences, further helping to identify evidence gaps and research priorities.

► An appraisal of the methodological quality of studies is not a requirement of a scoping review; however, a quality appraisal of the included research articles will be conducted to gather insights into the types, sources and quality of the evidence around the use of patient-reported outcomes in adult social care.

## INTRODUCTION

People with care and support needs often live with multiple and complex, long-term health conditions.[1] They are likely to experience greater levels of disability and have complex care and support needs.[2 3] An understanding of a person's own views about their health can help ensure care and support is provided in a way that is holistic, person-centred and responsive. Such insights are particularly relevant in the UK where a fragmented health and social care system make the coordination of care challenging.[4]

Individuals' perspectives about their health are most useful to care providers when these first-hand accounts are both relevant and reliably measured.[5] A patient-reported outcome (PRO) is 'a measurement of any aspect of a patient's health that comes directly from the patient without the interpretation of the patient's responses by anyone else'.[6] PROs include symptom severity of the impact of a disease or its treatment on a patient's health-related quality of life (including functioning and social and emotional well-being).[7] PROs are typically measured using self-report questionnaires known as patient-reported outcome measures (PROMs).

PROs are implemented widely across different levels of healthcare.[8] At an individual

level, PROs may be used to inform decision-making around treatment, monitor disease symptoms and treatment adherence and facilitate communication between the patient and their clinical team. At a systems level, PROs may be used to support benchmarking including service evaluation and commissioning of services, inform policymaking and support quality improvement initiatives. PROs are used as endpoints within clinical trials, support regulatory approval of new medicinal products and devices and inform clinical guidelines and health policy.[5 9 10] PROs are valued due to their capability to ensure that the patient's unique perspective is represented in the measurement of health and when capturing the effectiveness of healthcare interventions.

Despite their established use in healthcare, the role of PROs and PROMs in social care is less clear. Published reviews have tended to study PROMs within a specific context, health condition or population. For example, reviews have explored the use of PROMs in primary care and investigated stakeholders' perspectives regarding the use of PROMs with frail, older adults living at home.[11 12] Reviews have also been conducted that focus on the use of PROMs in end-of-life care and in the care of people with dementia and other long-term conditions.[13–16] To date, a review focussing on the use of PROMs in the broader context of adult social care has not been identified.

The Care Act (2014) places a duty on local authorities to ensure that the well-being of individuals is placed at the heart of care and support.[17] Commensurately, in social care, PROMs have focused on measuring the wider determinants of well-being (eg, personal dignity, control by an individual, participation in work or training and mental, physical and emotional health) and social care-related quality of life.[18] However, it has been suggested that health, as a component of well-being, may be more appropriately measured with health-related PROMs as a complement existing social care measures.[19 20] A better understanding of how health and social care-related PROMs are used in the delivery of care and support, by whom, and the barriers and facilitators to their utilisation may be considered increasingly relevant as decision-making around care and support becomes more person focussed and data driven.[21] A comprehensive understanding of PROs may be particularly pertinent as health, care and support services move towards greater integration.[22] Clarification on PROM use is especially timely in light of the COVID-19 pandemic, which has resulted in the increased use of telemedicine, virtual consultations and remote assessment and monitoring.[23 24] As such, PROMs have the potential to help people with care and support needs express their views about their health (including symptoms, functioning and quality of life) and well-being.[25]

Consistent with the remit of scoping reviews as a form of knowledge synthesis, the aim of this scoping review is to explore the existing evidence to understand the current state of knowledge on the use of PROs and PROMs in adult social care. To achieve this aim, the specific review objectives are: (1) to identify PROMs used by people to meet their own direct care and support needs, (2) to characterise how PROMs have been used in this context; and to chart the evidence relating to (3) their effectiveness, (4) barriers and facilitators to their implementation and (5) stakeholder views on their use. A subanalysis to explore the use of PROMs in the delivery of integrated health and social care will also be undertaken.[26]

## A shared language for PROs

The term 'patient-reported outcome' is a reflection of PROs' origins in medicine and healthcare. However, in the context of social care people with care and support needs rarely identify as patients, preferring to identify as service users or citizens.[27] In this review on the use of PROs within adult social care, the terms 'person-reported outcome' and 'person-reported outcome measure' will be used.

## METHODS AND ANALYSIS
### Protocol design, registration and reporting

Scoping reviews are conducted to understand the existing research on a given topic and often serve as a precursor to systematic reviews.[28] Specifically, scoping reviews seek to identify and map the available evidence and, in so doing, identify knowledge gaps and directions for future research.[29] Frequently, scoping reviews are complemented by stakeholder consultation exercises to ensure a holistic understanding of the topic that is informed by both the published evidence and lived experience.[30] This scoping review will use the methodology proposed by the Joanna Briggs Institute, supported by the original scoping framework proposed by Arksey and O'Malley and Peters et al.[30 31] The Preferred Reporting Items for Systematic Reviews and Meta-analyses (PRISMA) guidance for protocol development and scoping reviews and Joanna Briggs Institute (JBI) reporting checklists were consulted when preparing this protocol.[31–33] This review protocol is registered online with the Open Science Framework ( www.osf.io).

### Eligibility criteria
#### Participants

The review will consider sources of evidence that report on adults ($\geq$ 18 years of age) who are direct users of social care services or receive integrated health and social care. No limitations with regards to an individual's health condition(s) will be applied.

#### Concept

Sources of evidence that provide information on any PROM used in integrated or social care will be considered for inclusion in the review. Self-report instruments that are proxy-reported and carer-reported measures will be excluded.

#### Context

Sources of evidence reporting on a social care or integrated care intervention will be considered. An intervention may

be defined as an 'action where someone gets involved to improve a situation or prevent it getting worse' that is undertaken with, or on behalf of, a service user to promote independence, provide support, prevent harm and enable them to live their lives in accordance with their own wishes and beliefs.[34 35] Integrated care refers to coordinated health and social care that is planned and organised around the needs and preferences of the individual, their carers and family.[22 34] The review will exclude studies and other evidence sources set in primary and secondary care contexts.

## Types of sources

This scoping review will consider all sources of published evidence on the use of PROs and PROMs in adult social care. Sources of evidence will include primary research studies, websites, guidelines, government reports and policy documents. Expert opinions, editorials and studies reporting secondary data analysis (ie, systematic reviews) will also be included.

## Information sources

The searches will be conducted in three stages: (1) electronic database searching, (2) grey literature searching and (3) hand searching the reference lists and citations of included sources to identify further studies for inclusion. Electronic database selection will be based on recommendations for optimal database searching for sensitivity.[36]

The electronic databases Medline (Ovid), Embase (Ovid), PsychInfo (Ovid), HMIC (Ovid), Social Care Online (SCIE), ASSIA (Pro Quest) and Web of Science (Pro Quest) were searched on 29 September 2020 for the period 2010 to present date, with no language limitation. Sources published in languages other than English will be translated. Grey literature databases, Google and websites of government and third sector organisations will be searched to locate relevant sources. Hand searching of all of the included sources' reference lists will also be undertaken. Grey literature will be defined as 'work that is produced on all levels of government, academia, business and industry in print and electronic formats but which is not controlled by commercial publishers.[37] A 10-year date limit on searches was considered appropriate for locating relevant sources, reflecting the increased use of PROMs over the last decade, the introduction of the UK National PROMs programme, publication of the FDA's guidance on PROMs and the UK government's National Health Service (NHS) white paper outlining government's vision for greater integration of health and social care.[6 38 39]

## Search strategy

The Medline search strategy will be developed and then adapted for use with other databases. The search strategy will combine MeSH terms (and relevant synonyms) and the Boolean operators "AND" and "OR" for search strings. Four concept clusters will be included: (1) the concept of interest (ie, PROs and PROMs), (2) the population of adults who use social care or integrated care services

(3) contextual factors including type of integrated care/social care intervention and (4) PROM implementation (eg, barriers and facilitators). Terms were developed by the review team with support from an information science specialist. Service user partner and coinvestigator (PC) was consulted to identify specific search terms related to integrated care and social care. The grey literature search will take place after searches of the electronic databases have been completed to enable to study team to immerse themselves in the literature and come to better understand the evidence base. Hand searching of citations and reference lists of included documents will be undertaken at each stage of the search process. The Medline search strategy is presented in online supplemental file 1.

## Selection of sources of evidence

A team of reviewers (SEH, GT, CM, SCR) will conduct title/abstract and full-text screening of all records against prespecified eligibility criteria to ascertain suitability for inclusion in the review. All records will be screened independently by a minimum of two reviewers. Disagreements between reviewers will be resolved through discussion. If necessary, a third reviewer will be involved if consensus cannot be reached. The eligibility criteria will be pretested on a sample of abstracts to ensure that evidence sources discussing the use of PROs and PROMs in integrated/social care are captured. Records identified from electronic databases, the grey literature and hand searching will be imported to Endnote V.9.3.3 (www.endnote.com) reference management software, and duplicate records will be removed. All screening will be conducted within Endnote. Any retrieved documents deemed to meet the inclusion criteria will go forward for full-text review. For articles excluded during full-text screening, a reason for exclusion will be documented. Disagreements on article inclusion will be resolved through discussion by members of the review team until consensus is achieved. The study selection process will be documented in a PRISMA flow diagram.

## Critical appraisal of individual sources of evidence

Assessment of the methodological quality of included studies is not a requirement of scoping reviews.[31] The decision to undertake a quality appraisal will be governed by the number and type of sources identified by the searches. If feasible, a formal quality assessment will be undertaken using the Critical Appraisal Skills Programme checklists and the AACODS (Authority, Accuracy, Coverage, Objectivity, Date, Significance) checklist for quality appraisal of the grey literature.[40 41] Appraisals will be undertaken to characterise the overall quality of the evidence base and no study will be excluded on the basis of its methodological quality. Two reviewers will independently appraise studies for quality. Findings will be collated by study type and presented in tabular format along with a textual description of findings. To ensure transparency of reporting, interrater agreement and reliability will be

determined by calculating an intraclass coefficient (ICC) statistic, Cohen's κ and percentage agreement.[42 43]

## Data charting process

Data extraction, termed data charting in scoping reviews, aims to provide a 'logical and descriptive summary of results that align with the review questions'.[31] A standardised template will be developed in Excel and piloted prior to commencing data charting. Two reviewers will extract key characteristics of the data source and information relevant to answering the review questions independently. Discrepancies will be resolved through discussion. As a scoping review is an iterative process, the data extraction form will be refined and updated throughout the course of the review and in response to the emergent findings.

## Data items

The data items to be extracted for each data source comprise: (1) the publication title, (2) date of publication, (3) authors, (4) country (location of study), (5) study aims and objectives, (6) study design, (7) study setting, (8) whether the intervention is an integrated care intervention, (9) study population, (10) study data collection method, (11) data analysis method. Information on community and user involvement, study outcome(s), process and system-level outcomes in response to the PROM intervention and barriers/enablers to implementation will also be extracted. PROM-specific information will be charted separately and include a list of studies using each PROM and, if available, a description of the PROMs' characteristics, including target population, construct domains, number of items/subscales, mode of administration, recall period, response options, range of scores, scoring, language of the original publication and available translations.

## Data summary

A qualitative content analysis in scoping reviews is generally descriptive in nature.[31] Following data charting, findings from the included sources will be collated with analysis of common themes relating to the aims of the review. Given the anticipated diversity of the included data sources, the framework method, using NVivo (V.1.3) qualitative data analysis software and Microsoft Excel spreadsheets and data charting, will be used to capture key themes.[44 45] The framework method sits within the family of methods described as thematic analysis or content analysis. Its hallmark is the systematic construction of a matrix that facilitates comparison across and within cases (ie, data sources) to produce a thematic interpretation of the data. The framework method follows a structured format, whereby the reviewers will first become familiar with the data source. The data will be reviewed line-by-line and coded by applying a paraphrase or label. Coding will include a number of *a priori* deductive codes selected for their relevance to the aims of this scoping review to chart the evidence of PROM use in adult social care. The

reviewer will also deploy inductive coding by remaining open to the coding of new and unexpected concepts that may be emergent in the data. The first few data sources will be coded independently by two reviewers. Through consensus discussions, the reviewers will agree on a set of codes to be applied to all subsequent sources. Codes will be grouped into categories to create an analytical framework that will be used for indexing the remaining data sources. Finally, the coded data will be charted into a matrix using a spreadsheet. The characteristics of and differences between the data will be identified and a description of relationships between categories will be developed. Results will be presented using tables and diagrams accompanied by a narrative summary.[31]

## Piloting/calibration exercise

All phases of the scoping review will be pretested. The JBI recommended procedure for pilot testing will be used.[31] For screening, the titles and abstracts of 25 randomly selected articles will be screened by the review team, applying the *a priori* review eligibility criteria. Screening will commence once inter-rater agreement of 75% or greater is achieved. Inter-rater agreement that is less than 75% will prompt the study team to review and refine the inclusion criteria. This process of pilot testing will continue iteratively until the 75% agreement threshold is met.

The data charting template will be piloted by two reviewers to ensure the capture of data relevant to the aims and objective of the review. As the review progresses, the reviewers may wish to extract additional information of relevance to the review objectives; therefore, further revisions to the data chart template may be made iteratively as the review progresses.[31]

## Changes to the protocol

The exploratory nature and breadth of a scoping review mean that changes to the protocol may be required as new information becomes available and as the review progresses. Any protocol deviations or refinements, together with a rationale for these changes, will be documented clearly in the review report.

## Service user and public involvement

This scoping review was conceived and developed with community and user involvement. As coinvestigator and a person with lived experience, PC contributed to the development of the search strategies, the study protocol, consensus discussions and will review all study-related documents. A lay summary to facilitate dissemination of the review findings will be coproduced. In addition, this work will be discussed with the NIHR ARC West Midlands Long Term Conditions Theme 1 Patient and Public Involvement and Engagement Stakeholder Advisory Group. Meetings will be held on a quarterly basis throughout the review process. The Guidance for Reporting Involvement of Patients and the Public-2 short

form checklist will be used to document service user and public involvement.[46]

## Ethics and dissemination

Ethical approval is not required as this review is a retrospective review of publicly available evidence sources. The review findings will be disseminated via publication in a peer-reviewed journal, symposia and conference presentations. A dissemination strategy to ensure the review findings reach relevant stakeholders will be coproduced in partnership with members of the review team who are experts by experience.

## DISCUSSION

PROs can provide key data that enable health and social care practitioners to understand better a person's views about their health and well-being, allowing them to provide effective, efficient and compassionate care. This scoping review will be the first to provide an overview of the evidence around the use of PROs with adults who use social care services. Given the disparate nature of the evidence, the use of a scoping review methodology is appropriate as it enables the review to evolve in an iterative manner that is responsive to the evidence.

**Author affiliations**
[1]Centre for Patient Reported Outcomes Research (CPROR), University of Birmingham, Birmingham, UK
[2]Institute of Applied Health Research, University of Birmingham, Birmingham, UK
[3]National Institute for Health Research Applied Research Centre West Midlands, University of Birmingham, Birmingham, UK
[4]NIHR Birmingham Biomedical Research Centre, University of Birmingham, University of Birmingham, Birmingham, UK
[5]University Hospitals Birmingham NHS Foundation Trust, Birmingham, UK
[6]University of Warwick Warwick Medical School, Coventry, UK
[7]Birmingham Health Partners Centre for Regulatory Science and Innovation, Birmingham, UK
[8]Department of Social Work and Social Care, University of Birmingham, Birmingham, UK
[9]National Institute for Health Research Surgical Reconstruction and Microbiology Research Centre, Birmingham, UK

**Acknowledgements** The authors wish to thank Sue Bayliss, Information Specialist, University of Birmingham for her assistance in developing the search strategy.

**Contributors** SEH, MC, OLA, DSL, JG conceptualised the study and contributed to the study design. SEH and PC developed the search strategy. GMT, SCR, CM contributed to methods design. SEH drafted and edited the manuscript with MC, OLA, DSL, JG, PC, GMT, SCR, CM providing critical revisions to the manuscript and supplementary material. The final version was read and approved by all authors (SEH, MC, OLA, DSL, JG, PC, GMT, SCR, CM).

**Funding** This research is funded by the National Institute for Health Research (NIHR) Applied Research Collaboration (ARC) West Midlands. The views expressed are those of the author(s) and not necessarily those of the NIHR or the Department of Health and Social Care.

**Competing interests** SEH is supported by the National Institute of Health Research (NIHR) Applied Research Centre (ARC), West Midlands. SH is company director of Narra Consulting Limited and undertakes consultancy work for Cochlear Ltd. MC is a National Institute for Health Research (NIHR) Senior Investigator and receives funding from the National Institute for Health Research (NIHR) Birmingham Biomedical Research Centre, the NIHR Surgical Reconstruction and Microbiology Research Centre and NIHR ARC West Midlands at the University of Birmingham and University Hospitals Birmingham NHS Foundation Trust, Health Data Research UK, Innovate UK (part of UK Research and Innovation), Macmillan Cancer Support, UCB Pharma MC has received personal fees from Astellas, Takeda, Merck, Daiichi Sankyo, Glaukos, GSK and the Patient-Centered Outcomes Research Institute (PCORI) outside the submitted work. OLA is supported by the National Institute of Health Research (NIHR) Birmingham Biomedical Research Centre (BRC), Birmingham, West Midlands. OLA also receives funding from the Health Foundation and has received personal fees from Gilead Sciences. DL is supported by the NIHR ARC West Midlands and the NIHR Community Healthcare MedTech and IVD Cooperative, hosted by Oxford Health NHS Foundation Trust. JG is a non-executive director of University Hospitals Birmingham, adjunct professor at Curtin University, member of the leadership team for an ESRC large grant on sustainable adult social care, and member of the expert advisory group for West Midlands ARC. PC, GT, SCR, and CM have no conflicts of interest to declare.

**Patient consent for publication** Not required.

**Provenance and peer review** Not commissioned; externally peer reviewed.

**ORCID iDs**
Sarah E Hughes http://orcid.org/0000-0001-5656-1198
Samantha Cruz Rivera http://orcid.org/0000-0002-1566-6804
Christel McMullan http://orcid.org/0000-0002-0878-1513
Melanie Calvert http://orcid.org/0000-0002-1856-837X

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
