## [Reviewer comments · BMJ Open]

ARTICLE DETAILS

TITLE (PROVISIONAL)	Protocol for a scoping review exploring the use of patient reported outcomes in adult social care
AUTHORS	Hughes, Sarah; Aiyegbusi, Olalekan; Lasserson, Daniel; Collis, Philip; Cruz Rivera, Samantha; McMullan, Christel; Turner, Grace; Glasby, Jon; Calvert, Melanie

VERSION 1 – REVIEW

REVIEWER	Stacey Rand University of Kent, UK
REVIEW RETURNED	02-Dec-2020

GENERAL COMMENTS	p.7. lines 36-50. Here, it is implied that this scoping review will be limited to PROMs designed to capture HRQoL. However, the description of the methods and search strategy (Supplementary File 1) indicate that the scoping review is broader, i.e. it will also include PROMs designed to capture social care-quality of life or QoL, wellbeing outcomes, beyond health. It may be helpful to revise this section to reflect this. p.8 line 24. Will ‘users of social care services’ include (unpaid) carers? In England, carers may access and use social care services in their own right. However, it is not clear whether they will be included. (They do not seem to be included in the search strategy in Supplemental File 1?) p.11. line 36. It would be helpful to add a brief description of how the framework method will be applied. p.12. lines 32. “... this work will be discussed with the NIHR Long Term Conditions Theme 1 Patient and Public Involvement (PPI) Stakeholder Advisory Group at various points in the review process.” It would be helpful to state the (approx.) frequency or number of consultations.
--

REVIEWER	Juliette Malley London School of Economics and Political Science, UK
REVIEW RETURNED	07-Dec-2020

GENERAL COMMENTS	This study will undertake a scoping review of PROs in adult social care to explore the current state of knowledge on the use of PROs in social care. The researchers note that in the review “patient or person-reported outcome” (PRO) will refer specifically to health-related outcomes reported directly by an individual and will exclude measurements of a person’s experience of care or social care-related outcomes. They will identify not only which PROs are used in
--

	social care interventions, but how they have been applied, their effectiveness, barriers and facilitators to their implementation, and stakeholder views on the use of PROs in social care. In general the study is well-described, the methods seem appropriate and thorough, and the protocol is clearly written. I can see the value in this research and would be very keen to read the findings from the completed study. There are, however, four points which require some clarification:  1. It would be helpful to strengthen the rationale for the focus of the review on PROs for health-related outcomes in a social care context, as this focus sits somewhat uncomfortably with the direction of policy and practice in the UK. For social care services generally a much wider set of outcomes is considered to be relevant. Both policymakers and practitioners tend to focus on wellbeing or quality of life (see for instance the Care Act 2014 and Adult Social Care Outcomes Framework), as these are most relevant to the activities of social care. Indeed, NICE recommends the use of ASCOT or ICECAP, two much broader quality of life-oriented person-reported outcome measures, for assessing the effectiveness of social care interventions. While I accept both that many social care users will have underlying health conditions that drive their use of social care services, and that there is often significant health care input for these individuals, the primary purpose of social care interventions is to enable people to live as full a life as possible. Since this has consequences for the extent to which social care stakeholders see health-related outcomes as relevant, and therefore the extent to which they are likely to be used within social care, my view is that this point needs recognition and some discussion. 2. In the abstract it is important to clarify that the scoping review is looking solely at the use of health-related outcomes, as given the above point, I think some readers may have a broader interpretation. 3. What is meant by social care interventions is not expanded upon in the protocol (except in the search teams). It would be helpful to have this elaborated upon in the main body of the paper. Equally it would be helpful to have some understanding of what is meant by an integrated care intervention, as this could have a number of interpretations. 4. I am not completely clear on whether the segment of the study that is looking at stakeholder views on the use of PROs is part of this protocol or not. It is mentioned as an objective of the review (line 123, p5), but later in the protocol it is suggested that this is not part of this protocol (lines 299-305, p10). This section is not described in much detail. If this aspect is part of the study then it would be helpful to have more details about which stakeholders will be included, e.g. from which staff groups and social care settings. If this is not part of this protocol then it might be better to omit this element as one of the objectives.
--	---

VERSION 1 – AUTHOR RESPONSE

Reviewer 1

1. p.7. lines 36-50. Here, it is implied that this scoping review will be limited to PROMs designed to capture HRQoL. However, the description of the methods and search strategy (Supplementary File 1) indicate that the scoping review is broader, i.e. it will also include PROMs designed to capture social care-quality of life or QoL, wellbeing outcomes, beyond health. It may be helpful to revise this section

to reflect this.

Thank you, we have clarified the broader scope of the review in the manuscript with the following text:

“The Care Act (2014) places a duty upon Local Authorities to ensure that the wellbeing of individuals is placed at the heart of care and support.¹⁷ Commensurately, in social care, PROMs have focussed on measuring the wider determinants of well-being (e.g., personal dignity, control by an individual, participation in work or training, and mental, physical and emotional health) and social-care related quality of life.¹⁸ However, it has been suggested that health, as a component of well-being, may be more appropriately measured with health-related PROMs as a complement existing social care measures.¹⁹ ²⁰ A better understanding of how health and social care-related PROMs are utilised in the delivery of care and support, by whom, and the barriers and facilitators to their utilisation, may be considered increasingly relevant as decision-making around care and support becomes more person-focussed and data driven.²¹ A comprehensive understanding of PROs may be particularly pertinent as health, care and support services move towards greater integration.²² Clarification on PROM use is especially timely in light of the Covid-19 pandemic, which has resulted in the increased use of telemedicine, virtual consultations, and remote assessment and monitoring.²³ ²⁴ As such, PROMs have the potential to help people with care and support needs express their views about their health (including symptoms, functioning, and quality of life) and well-being.²⁵” (Lines 107-122, pp.4-5)

Also:

“This scoping review will consider all sources of published evidence on the use of PROs and PROMs in adult social care.” (Lines 172-173, p. 6)

2. p.8 line 24. Will ‘users of social care services’ include (unpaid) carers? In England, carers may access and use social care services in their own right. However, it is not clear whether they will be included. (They do not seem to be included in the search strategy in Supplemental File 1?)

Thank you for your comment. We acknowledge that many carers are ‘users of social care services’ in their own right and that a number of PROMs are available for use with this group. However, for purposes of this review, we wish to focus our inquiry on PROMs that could benefit people with care and support needs who are direct users of social care. An exploration of instruments designed for use with carers is important work and will be discussed in the context of future directions for research.

3. p.11. line 36. It would be helpful to add a brief description of how the framework method will be applied.

Thank you for your helpful comment. We have added the following description to the manuscript: “...the framework method, using NVivo (Version 1.3) qualitative data analysis software and Microsoft Excel spreadsheets, and data charting will be used to capture key themes.⁴⁴ ⁴⁵ The framework method sits within in the family of methods described as thematic analysis or content analysis. Its hallmark is the systematic construction of a matrix that facilitates comparison across and within cases (i.e., data sources) to produce a thematic interpretation of the data. The framework method follows a structured format whereby the reviewers will first become familiar with the data source. The data will be reviewed line-by-line and coded by applying a paraphrase or label. Coding will include a number of a priori deductive codes selected for their relevance to the aims of this scoping review to chart the evidence of PROM use in adult social care. The reviewer will also deploy inductive coding by remaining open to the coding of new and unexpected concepts that may be emergent in the data. The first few data sources will be coded independently by two reviewers. Through consensus discussions, the reviewers will agree on a set of codes to be applied to all subsequent sources. Codes will be grouped into categories to create an analytical framework that will be used for indexing the remaining

data sources. Lastly, the coded data will be charted into a matrix using a spreadsheet. The characteristics of and differences between the data will be identified and a description of relationships between categories will be developed....” (Lines 269-283, pp. 9-10).

4. p.12. lines 32. “... this work will be discussed with the NIHR Long Term Conditions Theme 1 Patient and Public Involvement (PPI) Stakeholder Advisory Group at various points in the review process.” It would be helpful to state the (approx.) frequency or number of consultations.

Thank you for your suggestion. We have amended the manuscript to include the following statement:

“In addition, this work will be discussed with the NIHR ARC West Midlands Long Term Conditions Theme 1 Patient and Public Involvement and Engagement (PPIE) Stakeholder Advisory Group. Meetings will be held on a quarterly basis throughout the review process.” (Lines 311-314, pp. 10-11)

Reviewer 2:

1. It would be helpful to strengthen the rationale for the focus of the review on PROs for health-related outcomes in a social care context, as this focus sits somewhat uncomfortably with the direction of policy and practice in the UK. For social care services generally a much wider set of outcomes is considered to be relevant. Both policymakers and practitioners tend to focus on wellbeing or quality of life (see for instance the Care Act 2014 and Adult Social Care Outcomes Framework), as these are most relevant to the activities of social care. Indeed, NICE recommends the use of ASCOT or ICECAP, two much broader quality of life-oriented person-reported outcome measures, for assessing the effectiveness of social care interventions. While I accept both that many social care users will have underlying health conditions that drive their use of social care services, and that there is often significant health care input for these individuals, the primary purpose of social care interventions is to enable people to live as full a life as possible. Since this has consequences for the extent to which social care stakeholders see health-related outcomes as relevant, and therefore the extent to which they are likely to be used within social care, my view is that this point needs recognition and some discussion. In the abstract it is important to clarify that the scoping review is looking solely at the use of health-related outcomes, as given the above point, I think some readers may have a broader interpretation.

Thank you, we have added the following to the manuscript to strengthen the rationale for the review:

“The Care Act (2014) places a duty upon Local Authorities to ensure that the wellbeing of individuals is placed at the heart of care and support.¹⁷ Commensurately, in social care, PROMs have focussed on measuring the wider determinants of well-being (e.g., personal dignity, control by an individual, participation in work or training, and mental, physical and emotional health) and social-care related quality of life.¹⁸ However, it has been suggested that health, as a component of well-being, may be more appropriately measured with health-related PROMs as a complement existing social care measures.¹⁹ ²⁰ A better understanding of how health and social care-related PROMs are utilised in the delivery of care and support, by whom, and the barriers and facilitators to their utilisation, may be considered increasingly relevant as decision-making around care and support becomes more person-focussed and data driven.²¹ A comprehensive understanding of PROs may be particularly pertinent as health, care and support services move towards greater integration.²² Clarification on PROM use is especially timely in light of the Covid-19 pandemic, which has resulted in the increased use of telemedicine, virtual consultations, and remote assessment and monitoring.²³ ²⁴ As such, PROMs have the potential to help people with care and support needs express their views about their health (including symptoms, functioning, and quality of life) and well-being.²⁵” (Lines 107-122, pp.4-5)

2. What is meant by social care interventions is not expanded upon in the protocol (except in the search teams). It would be helpful to have this elaborated upon in the main body of the paper.

We have defined social care interventions in the manuscript and have added the following text:

Sources of evidence reporting on a social care or integrated care intervention will be considered. An intervention may be defined as an “action where someone gets involved to improve a situation or prevent it getting worse” that is undertaken with, or on behalf of, a service user to promote independence, provide support, prevent harm and enable them to live their lives in accordance with their own wishes and beliefs.^{34 35} Integrated care refers to coordinated health and social care that is planned and organised around the needs and preferences of the individual, their carers and family.^{22 34} (Lines 163-168, p.6)

3. Equally it would be helpful to have some understanding of what is meant by an integrated care intervention, as this could have a number of interpretations.

For the purposes of this review, we have focussed on models of integrated care coordinated across health and social care services. We have clarified the meaning of integrated care in the manuscript by adding the following description:

“Integrated care refers to coordinated health and social care that is planned and organised around the needs and preferences of the individual, their carers and family.^{23 35} (Lines 167-169, p. 6)

4. I am not completely clear on whether the segment of the study that is looking at stakeholder views on the use of PROs is part of this protocol or not. It is mentioned as an objective of the review (line 123, p5), but later in the protocol it is suggested that this is not part of this protocol (lines 299-305, p10). This section is not described in much detail. If this aspects is part of the study then it would be helpful to have more details about which stakeholders will be included, e.g. from which staff groups and social care settings. If this is not part of this protocol then it might be better to omit this element as one of the objectives.

Thank you for your comment. The protocol for the stakeholder consultation is still under development; therefore, we have removed this element of the review from the protocol, as suggested.